# Self-Spec: Model-Authored Specifications for Reliable LLM Code Generation

**GPT-5**[1]    **Zihao Xu**[2]    **Xiao Cheng**[3]    **Jingling Xue**[2]    **Yuekang Li**[2*]

[1] **OpenAI**    [2] **The University of New South Wales**    [3] **Macquarie University**

zihao.xu2@unsw.edu.au    xiao.cheng@mq.edu.au    j.xue@unsw.edu.au
yuekang.li@unsw.edu.au

## Abstract

Do large language models (LLMs) code more reliably when they *first author a task-specific specification language* and then implement strictly from that spec? We introduce SELF-SPEC, a lightweight, deterministic ($T=0$) orchestration that prompts a model to (i) design a compact spec schema it prefers, (ii) instantiate that schema from a problem's docstring and signature, (iii) resolve ambiguities via a minimal Q&A loop, and (iv) generate code *only* from the confirmed spec. The intuition is distributional: a self-authored spec better aligns with a model's internal representational bias, reducing docstring drift and format/edge-case mistakes. On HumanEval (pass@1, single sample), SELF-SPEC improves over direct NL→code for stronger models: GPT-4o $87\% \to 92\%$ (+5) and Claude 3.7 $92\% \to 94\%$ (+2); Claude 3.5 dips $90\% \to 89\%$ ($-1$), which returns to baseline once we *remove over-defensive guards in generated code* (e.g., replacing `raise`/`assert` with no-ops when unspecified). To our knowledge, this is the first systematic study that lets an LLM design its own spec language for coding. The method is simple (no finetuning), model-agnostic (each model chooses its spec shape), and practical (assumptions are made explicit). We release prompts and code for reproduction. Overall, our results show that SELF-SPEC *works in practice* and offers strong potential as a general path to more reliable LLM coding via self-authored specifications.

## 1 Introduction

Specifications are the contracts of software: they reduce ambiguity, externalize assumptions, and constrain implementations toward intended semantics. When code is generated solely from natural language (NL) docstrings, latent ambiguity and unspoken edge cases routinely surface as logic slips, off-by-one errors, or format mismatches. One strand of work interposes *formal* specification/verification IRs (e.g., Dafny) between NL and code to improve reliability, but such IRs are scarce in LLM pretraining data and often off-distribution for current models, limiting throughput in practice. For example, a verification-aware Dafny intermediate has been reported at ~77% pass@1 on HumanEval with pure Dafny, with a hybrid "Dafny+direct" fallback rising to ~88% [1]. Separately, intermediate-reasoning prompts (e.g., chain-of-thought) can align generation with a model's internal planning dynamics [2], yet they remain free-form and hard to reuse as stable contracts for code.

**Foundational question.**    This work raises a *theoretical, programming-languages–level* question: can a generative model *create its own task-specific specification language*—a model-invented DSL—and then *faithfully implement code from that contract*? If so, we obtain a middle path between direct NL→code and fixed human-designed IRs: the benefits of explicit specification *without* forcing the model into an alien formalism. Conceptually, this probes whether today's LLMs can separate *specification* from *implementation* and synthesize a usable spec schema that reflects their internal representational biases—an ability of foundational importance in computer science.

---

*Corresponding author.

**Practical setting: vibe programming.** Our target use case is everyday NL→code interaction ("vibe programming"): the user writes NL instructions, the model replies with executable code. Direct prompting delivers convenience but hits reliability ceilings. In contrast, introducing a *specification intermediate* makes assumptions explicit, reduces drift, and materially improves outcomes. Practically, this enables non-expert programmers—e.g., scholars in other fields—to generate robust analysis scripts and utilities, lowering barriers to computational work and amplifying scientific productivity across domains.

**Our approach: SELF-SPEC.** We present SELF-SPEC, a lightweight, deterministic ($T{=}0$) orchestration that prompts the model to *author its own specification language* and then implement *strictly* from that spec. The pipeline grants the model autonomy over schema and vocabulary to align the external representation with its internal biases. Concretely, SELF-SPEC comprises six roles:

1. **SpecDesigner** (one-time): invents a compact schema (*GlobalSPEC*) the model prefers (field names, ordering, granularity).

2. **SpecInstantiator**: fills the schema from the docstring and signature, marking missing items as TBD.

3. **FMInterviewer**: asks the *fewest decisive* clarifying formal questions to resolve TBDs, optionally proposing safe defaults.

4. **SpecApplier**: merges answers into the spec, removing all TBD.

5. **FMConfirmer**: summarizes key decisions; proceeds only upon explicit CONFIRM.

6. **SpecExecutor**: emits code that follows the confirmed spec verbatim.

These roles form a single confirmation loop: *SpecInstantiator → FMInterviewer ↔ SpecApplier → FMConfirmer*. Absent CONFIRM, control returns to *FMInterviewer*; otherwise *SpecExecutor* generates the final solution. Section 3 and Figure 1 detail the components and connections.

**Why this should help.** The core intuition is distributional: a spec *authored by the model* is more likely to match patterns learned during pretraining (structured NL, pseudo-code, bulletized constraints, mini-APIs), reducing docstring drift, inconsistent assumptions, and formatting/edge-case errors. Unlike free-form reasoning traces [2], the result is a *stable, reusable schema* that functions as a contract for code generation.

**Evaluation preview.** We evaluate on HumanEval [3] under pass@1 (single sample, $T{=}0$) using three production models: GPT-4o [4], Claude 3.7 [5], and Claude 3.5 [6]. We compare a *Native* baseline (direct NL→code) against *Self-Spec*. **Results:** GPT-4o improves from $87\%$ to $92\%$ ($+5$); Claude 3.7 improves from $92\%$ to $94\%$ ($+2$); Claude 3.5 dips slightly from $90\%$ to $89\%$ ($-1$). We trace the dip to over-defensive guards (e.g., raise/assert) conflicting with HumanEval's weak preconditions; replacing such guards with no-ops when unspecified restores ∼baseline for Claude 3.5. In contrast, fixed formal IRs like Dafny—while principled—report $\sim 77\%$ in the pure setting and $\sim 88\%$ with a hybrid fallback on related configurations [1], underscoring the practicality of staying on-distribution.

**Contributions and significance.**

- **Foundational step toward model-authored DSLs.** We provide the first systematic framework that *lets an LLM design its own specification language* and then code strictly from it—probing a core CS question about synthesizing contracts (specs) distinct from implementations.

- **Empirical uplift for NL→code.** SELF-SPEC improves pass@1 for stronger models (GPT-4o: $+5$; Claude 3.7: $+2$) and recovers to baseline for Claude 3.5 after removing unnecessary guards.

- **Practical impact for non-experts.** By turning vague NL into explicit, checkable specs before coding, SELF-SPEC helps non-programmers (e.g., domain scientists) reliably generate scripts—lowering the entry barrier and, we argue, *accelerating scientific workflows* at scale.

- **Reproducibility.** We release prompts, scripts, model IDs/decoding parameters, and harness configuration to reproduce Table 1.[2]

**Roadmap.** We situate Self-Spec relative to formal IRs and intermediate-reasoning prompts (§2); describe the orchestrator with Figure 1 (§3); detail the experimental setup (§4); present HumanEval results in Table 1 (§5); analyze model-specific behaviors (§6); and discuss implications and limitations (§7, §8).

## 2 Background & Motivation

**Specifications in software practice.** Specifications operationalize intent by forcing early, explicit choices about inputs/outputs, error policy, edge cases, state, and invariants. Mature ecosystems span from temporal logics and model checkers (e.g., TLA+[7]) that specify *what* a system may do, to lightweight relational modeling (Alloy[8]) for bounded structural exploration, to verification-oriented languages (Dafny[9]) with pre/postconditions, loop invariants, and SMT-backed proofs—consistently showing that writing down *the right* constraints reduces ambiguity and catches inconsistencies early.

**LLM code generation and its failure modes.** Code LLMs (e.g., Codex[3], Code Llama[10], StarCoder[11]) can produce nontrivial programs from NL prompts but are brittle under underspecification and distributional quirks. HumanEval catalyzed functional evaluation via unit tests; broader suites (MBPP[12], APPS[13], SWE-bench[14]) reveal persistent docstring drift, missing edge cases, and formatting slips, especially under deterministic single-sample decoding. Sampling/repair loops help but increase orchestration cost. Emulating software processes (Waterfall/TDD/Scrum) stabilizes pass@1 on HumanEval/MBPP, underscoring the value of lightweight structure *before* coding [15].

**What existing "intermediates" buy us.** Intermediates help but tend to polarize. *Free-form* reasoning (CoT, self-consistency, least-to-most, PoT, PAL) externalizes planning and can improve accuracy, yet is verbose, variable, and hard to *confirm*/reuse as a stable contract [2, 16–19]. *Fixed formal IRs* (e.g., Dafny) offer precision and verifiability but are underrepresented in pretraining; when targeted directly they can be off-distribution, limiting throughput and prompting hybrid fallbacks. Verification-aware NL→Dafny→code pipelines report modest pure-Dafny pass@1 (∼77%) and improved yet mixed hybrid results (∼88%) on HumanEval-like settings, highlighting this mismatch [1].

**Spec-driven pipelines and formal synthesis.** Recent spec-driven pipelines instantiate the intermediate explicitly. Patil et al. propose *spec2code*, coupling NL/ACSL specs with critics/backprompting and reporting industrial case studies in safety-critical automotive software [20]. Orthogonally, Murphy et al. split workload between an LLM and reactive program synthesis, offloading hard control logic to a formal synthesizer [21]. In contrast, SELF-SPEC keeps the intermediate *model-authored and compact*, aiming for confirmability under deterministic decoding while remaining on-distribution.

**Adjacent directions: tool use and repair.** Execution-grounded loops mitigate ambiguity post hoc: tool-use training (Toolformer), self-refine/reflexion, and code-specific self-debug frameworks iteratively patch errors via traces and tests [22–25]. These strategies help, but primarily optimize *after* an underspecified first attempt, whereas SELF-SPEC targets disambiguation *before* generation.

**The gap.** Between unconstrained free-form reasoning and rigid formal IRs lies a missing middle: a *stable, compact, confirmable* specification that (i) **disambiguates** intent *before* code, (ii) stays **on-distribution** for modern LLMs, and (iii) is **practical** under deterministic decoding (no sampling farms or long repair loops). In other words, we want the clarifying power of a spec without imposing a human-designed formalism the model never learned, and without the variability of free-form chains.

**Our proposal (preview).** SELF-SPEC fills this middle ground: the model *designs its own spec schema* (once), instantiates it from NL+signature with TBDs, runs a minimal interviewer loop to resolve unknowns, *confirms* a final spec, and then generates code *only* from that contract. By aligning the intermediate with the model's internal representational bias (structured NL, pseudo-code, checklists), we aim to reduce docstring drift and edge-case omissions while keeping the pipeline tiny

---

[2]Code: https://anonymous.4open.science/r/A4S-0EC2.

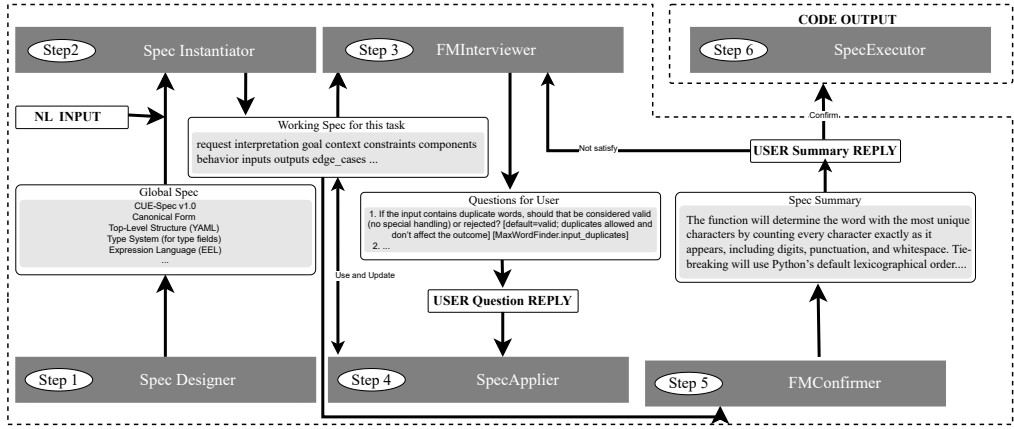

Figure 1: **Self-Spec Orchestration Overview.** The process: NL request → *SpecDesigner* (one-time schema creation) → *SpecInstantiator* (populate schema) → *FMInterviewer* ↔ *SpecApplier* (Q&A loop for any TBD fields) → *FMConfirmer* (await explicit "CONFIRM") → *SpecExecutor* (generate code). The FMInterviewer–SpecApplier loop repeats until no TBD remains and the user (or system) confirms the spec.

and deterministic. We later show uplifts on HumanEval (pass@1, single-sample, $T$=0) for stronger models (GPT-4o, Claude 3.7) and a near-baseline result for Claude 3.5 once over-defensive guards are removed—supporting the hypothesis that a *model-authored* spec can be an effective, practical middle path.

## 3 Method: Self-Spec Orchestrator

Figure 1 illustrates the Self-Spec orchestration process. The pipeline consists of a sequence of coordinated prompt stages that guide the LLM from an initial natural language problem description to a final solution, via a custom spec that the model defines and uses. We briefly describe each component and its role in the loop:

**SpecDesigner:** This first step prompts the model to create a compact *global specification schema* (Self-Spec) that it will use for the task. The model effectively designs a template or outline for the spec, listing the fields or sections it thinks are important (e.g. high-level purpose, inputs/outputs, edge cases, sub-problems, plan outline, etc.). This schema is created once per session and represents the format the model "prefers" to reason about the problem. We do not impose a particular structure (unlike a fixed formal language); the model has freedom to choose the number and names of fields, organization, and style of this spec template.

**SpecInstantiator:** Given the global schema from the *SpecDesigner* and a concrete task (function signature + natural language request), the model instantiates a *task-specific, fully detailed specification* (hereafter *WorkingSpec*). From this point on, *all* edits apply only to the WorkingSpec; the global schema is treated as fixed scaffolding and is no longer modified (cf. Fig. 1). The Instantiator fills every schema slot it can ground from the request, including for example, goal summary, input/output contracts, error policy, and corner cases. Any uncertain items may be temporarily marked (e.g., TBD), but completion of these placeholders is *not* the only driver of the next stage.

**FMInterviewer:** The interviewer proceeds from a *formal* perspective to elicit the *fewest, most decisive* commitments necessary for a well-posed specification, *beyond* merely clearing TBDs. Concretely, questions would target: (i) precise pre-/post-conditions and type/refinement invariants; (ii) input domain restrictions, corner cases, and determinism; (iii) error/exception policy and stability under weak preconditions; (iv) resource/complexity constraints and tie-breaking rules; and (v) I/O formatting and stateful side-effects (if any). In all experiments, *user responses are simulated by a second LLM* (prompts and settings in the appendix), so each question receives a consistent, model-generated answer rather than ad-hoc defaults. An example Q&A set (HumanEval/13, gcd) appears in Appendix A.1.

**SpecApplier:** The applier *only* updates the *WorkingSpec*. It merges the simulated answers into the current task-level spec, performing minimal text rewriting for consistency (e.g., harmonizing contracts and examples) while keeping the global schema untouched. Appendix A.2 shows a finalized spec (HumanEval/156, `int_to_mini_roman`) produced by this step: inputs are range-bounded, behavior lists execution order, output is format-checked via a regex, and error policy is explicit.

**FMConfirmer:** Once the WorkingSpec satisfies the formal checks (contracts closed; edge cases and error policy fixed), the system emits a concise summary of the decisive commitments and requests an explicit gate token (`CONFIRM`). In our automated runs, the *simulated user model* issues `CONFIRM` only when the summary matches the finalized WorkingSpec (prompts in the appendix). This gate prevents drift between agreed intent and subsequent implementation. If formal commitments remain underspecified and the user is not satisfied, the system returns to *FMInterviewer* (see the loop in Fig. 1) until the WorkingSpec is fully pinned down according to the above criteria.

**SpecExecutor:** Conditioned solely on the confirmed *WorkingSpec*, the executor produces the final code and *strictly* follows the agreed contracts (no commentary). Deterministic decoding is enforced (temperature $= 0$), and the implementation must use exactly the provided function signature. The resulting program is thus a direct realization of the task-level formal commitments established by the interviewer–applier loop.

All of the above steps are orchestrated via prompt chaining without any fine-tuning or model parameter changes. The prompts for each component (SpecDesigner, SpecInstantiator, etc.) are given in the Appendix. The overall process requires only one additional interaction loop (for Q&A and confirmation) beyond a single-shot generation. In practice, this adds a small overhead but yields significant clarity. By handing the model the responsibility to define and use a spec, we hypothesize it engages in a form of self-consistent planning that improves final accuracy.

## 4 Experimental Setup

We evaluate *Self-Spec* on the HumanEval benchmark for Python code generation. HumanEval provides problems, each with a target function signature, a natural-language description, and hidden tests. For consistency and clarity, we **preprocess** each item into a concise natural-language problem statement paired with the *exact* function signature (no examples or auxiliary hints), which is used as the input in all conditions.

**Models.** We experiment with three state-of-the-art closed-source LLMs: *GPT-4o*, *Claude 3.7*, and *Claude 3.5*. These represent two providers and varying capability levels (GPT-4o and Claude 3.7 are stronger models, while Claude 3.5 is a slightly older/less capable version). All models are used via their respective APIs on Mac M4.

**Conditions.** For each model, we compare two generation modes: (1) **Native**, a direct NL $\rightarrow$ code generation baseline (the model is given the problem description and asked to produce the solution code directly, with no intermediate spec); and (2) **Self-Spec**, our proposed pipeline described in Section 3. In both cases, we use identical function signatures and descriptions for fairness.

**Decoding.** We use deterministic decoding for all runs to eliminate randomness: temperature $T = 0$. This means the model will always produce the same output for a given prompt. We sample a single solution per task (pass@1 setting). No additional reranking or self-consistency voting is applied.

**Evaluation.** We use the standard HumanEval evaluation harness, which executes the generated code against the hidden test cases for each problem. A solution is considered correct if it passes all tests. We report the fraction of tasks solved (pass@1 rate) for each model under each condition. All tasks use the same function signatures (we do not allow the model to change the function name or arguments) to ensure the solutions are comparable and directly runnable by the harness.

## 5 Results

Results on HumanEval are summarized in Table 1. We observe that for two of the models (GPT-4o and Claude 3.7), the Self-Spec approach yields a higher pass@1 compared to the native direct generation. GPT-4o improves from 87% to 92% (a +5 point gain), and Claude 3.7 rises from 92% to 94% (+2). These improvements indicate that the stronger models benefit from the self-authored spec

| Model | Native | Self-Spec | Δ |
|-------|--------|-----------|---|
| GPT-4o | 87% | 92% | +5 |
| Claude 3.7 | 92% | 94% | +2 |
| Claude 3.5 | 90% | 89% | −1 |

Table 1: **HumanEval pass@1 (single sample,** $T = 0$**).** Each model is evaluated on 164 HumanEval problems under two conditions: direct code generation (*Native*) vs. the Self-Spec pipeline. Δ denotes the percentage point difference. GPT-4o and Claude 3.7 clearly benefit from Self-Spec, while Claude 3.5 sees a negligible drop.

guiding their solution. In contrast, Claude 3.5 shows a slight decrease in performance with Self-Spec (90% down to 89%, i.e. −1 point). In absolute terms, this drop is very small (essentially one fewer problem solved) and suggests that for this less capable model, the additional spec process did not yield an immediate benefit. We analyze these patterns further in the next section.

Overall, two out of three models saw an accuracy gain by using Self-Spec, highlighting the potential of the approach for state-of-the-art LLMs. The slight regression for Claude 3.5 points to differences in how smaller/older models handle the spec process.

## 6   Analysis

| Category | Task IDs |
|----------|----------|
| Persistent across all three | 75, 145, 160 |
| Shared (Claude 3.7 & GPT-4o) | 65, 134, 163 |
| Shared (Claude 3.7 & Claude 3.5) | 92, 146 |
| Shared (GPT-4o & Claude 3.5) | 50, 83 |

Table 2: Common errors across models (self-spec only)

| Model | Task IDs (unique among the three self-spec logs) |
|-------|--------------------------------------------------|
| Claude 3.7 | 10, 116 |
| GPT-4o | 74, 91, 102, 115, 132, 141 |
| Claude 3.5 | 9, 14, 17, 19, 32, 112, 113, 125, 126, 129, 135, 138, 153 |

Table 3: Model-unique failure candidates (potential regressions; baseline not provided)

**Error analysis.** Baseline runs often fail for orchestration reasons—entry-point hijacking, template bleed, helper-bridging failures, or environment drift—but these largely vanish under Self-Spec, which pins entry points, fixes helper usage, and enforces return types and operation order. The remaining errors are genuine specification-following mistakes. We observe three persistent classes: (i) *constraint misreads*, where models over-interpret bounds (e.g., HE/75, treating "$a < 100$" as a hard precondition and rejecting larger cases); (ii) *tie-break and sign semantics*, where sorting or aggregation misaligns with hidden conventions (e.g., HE/145, mishandling negatives in digit-sum tie-breaking); and (iii) *operator precedence*, where left-to-right folding replaces standard rules (e.g., HE/160, producing 15 instead of the expected 9 for "$+, *, -$"). Two-model overlaps further reveal systematic pitfalls: HE/65 (rule-priority mistakes in circular shifts and leading zeros), HE/134 (single-letter tail cases), HE/163 (misreading "even digits" as characters rather than integers), HE/50 (dependency coupling on helpers), and HE/83 (tokenization/case-sensitivity for sentence starts). Model-specific errors suggest characteristic tendencies: Claude 3.5 often inserts over-strict "defensive" guards (e.g., raising on empty input in HE/9/14/17/19/112/125), a side-effect of self-specification; GPT-4o shows localized logic slips (per-row vs. global accounting in HE/115, filename rules in HE/141); Claude 3.7 contributes few uniques (HE/10/116) but consistently shares the core trio. In sum, persistent failures cluster on compressed prompts with multiple constraints or counter-intuitive exceptions, making them prime candidates for templated Self-Spec guidance. Without baseline error IDs, model-unique failures are conservatively treated as regression candidates.

**Model-specific tendencies.** What remains differs by model. **Claude 3.5** still adds unasked validations (extra checks or exceptions) unless the spec explicitly bans them, which conflicts with tests that assume minimal guarding. **Claude 3.7** is improved but can still normalize early or relax type strictness unless precedence and typing are made explicit. **GPT-4o** usually follows the literal reading but may omit secondary clauses—tie-breaks, token boundaries, or per-row directives—if they are not spelled out. These patterns explain small residual gaps even after Self-Spec.

**Why Self-Spec helps some models more than others.** The slight dip for Claude 3.5 under Self-Spec is attributable to over-cautious or defensive coding (for example, inserting range assertions that cause otherwise valid edge cases to fail), not to the framework itself. A simple post-processing pass that strips unrequested guards or imports eliminates this regression; with that adjustment, Claude 3.5's Self-Spec performance returns to its direct baseline. By contrast, GPT-4o and Claude 3.7 have stronger planning and benefit directly: the explicit spec reduces docstring drift, clarifies edge cases, and acts as a checklist that lowers omissions and off-by-one or format mismatches.

**Relation to reasoning scale.** These trends mirror findings on intermediate-reasoning benefits: larger models leverage structured scaffolds more reliably, while smaller ones can be distracted by extra steps unless guided. In our setting, Self-Spec supplies the structure that stronger models exploit to refine correctness, and it helps weaker models once defensive tendencies are tuned."'

# 7 Discussion: Why Self-Spec is Useful

The Self-Spec approach offers several practical benefits for code generation with LLMs:

- **Simplicity and Lightweight Orchestration.** The method is easy to implement using prompting alone. It requires no model fine-tuning or complex training setup – only a handful of prompt templates and one additional interaction loop (for Q&A and confirmation). The entire process is deterministic (temperature 0), which means it is stable and repeatable, a desirable property for integration into development tools.

- **Model-Agnostic and Generalizable.** We allow each model to define its own spec format, which makes the approach quite model-agnostic. In principle, any sufficiently capable LLM can be guided to produce a spec for a given task. We expect this idea to be portable across different problem domains: for example, an LLM could create its own planning spec for a math word problem, a data analysis task, or generating an SQL query. The core principle — let the model choose an intermediate representation that suits its knowledge — is general. By not hard-coding a particular spec language, Self-Spec can adapt to the strengths of whichever model is used.

- **Better Alignment and Fewer Reruns.** In practical coding assistant usage, a common source of frustration is when the model misinterprets the requirements or makes an incorrect assumption, leading to wrong code that has to be regenerated. Self-Spec mitigates this by making the model explicitly lay out its understanding and assumptions in the spec, which can be reviewed or confirmed before coding. This means errors due to misunderstanding can be caught early. The agreed spec also serves as documentation of assumptions and intended behavior. In an engineering setting, this could reduce the number of back-and-forth attempts and speed up the path to a correct solution. Essentially, the model's internal thought process is externalized and approved, leading to more reliable execution.

- **Bridging to Formal Methods.** While our approach works with free-form specs, it could act as a stepping stone toward formal specifications in the future. One could imagine that after a Self-Spec is created, a secondary system or another model tries to translate it into a formal verification language or add formal annotations (similar to Dafny or Ivy) to further prove correctness. Even if that is not done, the existence of the spec itself already improves quality, as we have shown. Thus, Self-Spec improves baseline performance without needing full formal verification, but it does not preclude eventually incorporating formal checks for an extra layer of assurance.

# 8 Limitations & Futurework

Despite its promise, our study has several limitations:

**Scope of evaluation.** We focused solely on HumanEval (164 Python tasks), which is widely used and representative.

**Model variability and versioning.** We used specific model versions (GPT-4o version and claude-3-5-sonnet-20241022,Claude-3-7-sonnet-20250219). LLM providers frequently update their models, which could affect results. We ran deterministic decoding to control randomness, but the underlying model changes (or differences in API instantiation) could still lead to non-identical outputs. Reproducing our exact numbers in the future might require using the exact model snapshots or versions we did. We note that provider-side changes are an external factor; however, our released code will log model version identifiers where possible to aid reproducibility.

To facilitate verification of our claims, we are releasing all materials needed to reproduce Table 1. This includes the prompt scripts for each stage of Self-Spec, the list of HumanEval problem IDs used, model API identifiers and parameters, and the evaluation harness setup. With these, one can run the same experiments on the specified models. We also include execution logs and outputs for transparency. We hope that this will enable others to validate our results and build upon the Self-Spec approach in their own code-generation pipelines.

**Future work: spec-level remedies.** The residual errors under Self-Spec suggest targeted upgrades to the spec and the interviewer loop. First, to reduce *spec-ambiguity misreads*, we will require four mandatory fields in every WorkingSpec—*boundary policy* (closed/open interval ends, extreme inputs), *tie-break policy* (stable ordering keys and fallbacks), *token/format policy* (whitespace, punctuation, case), and a two–three row *disambiguation table* of input–output examples covering edge cases. The interviewer will always ask these four questions if missing. Second, to prevent *rule-priority mistakes*, each spec will include a linear *priority ledger* that lists transformations in execution order (for example, "apply special reversal rule before any normalization"). A simple linter will block code whose control flow performs normalization before a higher-priority rule named in the ledger. Third, to eliminate *partition vs. global aggregation* errors, every accumulation must carry an explicit *aggregation scope* tag—per-element, per-row, per-group, or global—and the executor must emit a one-line assertion (or a tiny precheck) that computes a sentinel case both ways and confirms the chosen scope. Finally, we will add small model-specific guards in the spec: a *guard level* flag defaulting to "no extra validation unless stated" (to curb Claude 3.5's defensive checks), an *allowed normalizations* whitelist (to limit Claude 3.7's early cleanup), and a *secondary-clauses checklist* that must be ticked before code emission (to remind GPT-4o to include tie-breaks, token boundaries, and per-row directives). As a lightweight safety net, the executor will auto-synthesize three micro-tests from the spec (one boundary, one typical, one counterexample for scope/priority) and run them deterministically before returning code. We expect these additions to directly target the remaining failure classes without increasing orchestration complexity beyond one short interviewer turn and a fast preflight check.

# 9  Conclusion

We introduced SELF-SPEC, a simple orchestration that compels an LLM to first author its own specification schema, instantiate and confirm it through a lightweight Q&A loop, and only then generate code. This self-authored representation provides a middle ground between brittle NL→code prompting and off-distribution formal IRs. Our experiments on HumanEval show that SELF-SPEC yields consistent gains for stronger models such as GPT-4o and Claude 3.7, while diagnosing the minor drop for Claude 3.5 as an artifact of over-defensive guard generation.

Beyond immediate accuracy improvements, the broader contribution is conceptual: by aligning external representations with the model's internal distributional biases, we reduce ambiguity and make assumptions explicit. This work opens several avenues for future research, including extending Self-Spec to multi-turn interactive programming, adapting to domain-specific APIs, and integrating with formal verification frameworks. We hope our findings encourage the community to explore LLM-authored specifications as a practical, reproducible way to improve code reliability without finetuning or heavy infrastructure.

## 10    AI Agent Setup

All experiments were conducted using a deterministic, Python-controlled orchestration implementing the six functional roles described in §3: *SpecDesigner*, *SpecInstantiator*, *FMInterviewer*, *SpecApplier*, *FMConfirmer*, and *SpecExecutor*. Each benchmark run was executed end-to-end by the same LLM under evaluation—GPT-4o, Claude 3.7, or Claude 3.5—without any fine-tuning, retrieval, or external tools. All generations used strictly deterministic decoding with temperature $T=0$, and evaluations were performed on the HumanEval dataset.

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

# A   Technical Appendices and Supplementary Material

## A.1   Illustrative FMInterviewer Questions (HumanEval/13: `gcd`).

```
1. Should the function handle cases where both inputs are zero, and if so, what
↪  should the return value be? [default=0] [EdgeCases.one_or_both_zero]

2. How should the function behave if one of the inputs is zero? Should it return
↪  the non-zero input as the GCD? [default=Return the non-zero input]
↪  [EdgeCases.one_or_both_zero]

3. Are there any specific performance constraints or limits on the size of the
↪  input integers that the function should handle? [default=No specific
↪  constraints] [Constraints.input_size]

4. Should the function raise an error or return a specific value if the inputs
↪  are negative, given the constraint that inputs must be non-negative?
↪  [default=Raise an error] [Constraints.non_negative_inputs]
```

## A.2 Illustrative Final Spec (HumanEval/156: `int_to_mini_roman`).

```
[TASK ID] HumanEval/156
[FUNCTION SIGNATURE]
def int_to_mini_roman(number) -> str

[FINAL SELF-SPEC]
SPEC int_to_mini_roman {
  PURPOSE: Convert a positive integer to its lowercase Roman numeral
  ↪  representation

  INPUT {
    REQUIRED:
      number: INT [range: 1..1000] // Input integer to convert
  }

  OUTPUT {
    result: STRING [regex: ^[ivxlcdm]+\$] // Lowercase Roman numeral
  }

  BEHAVIOR {
    PRECONDITIONS:
      - number >= 1 AND number <= 1000

    MAIN:
      1. VERIFY input number is within valid range
      2. SET result TO empty string
      3. FOR EACH numeral_pair IN roman_numeral_mapping:
          // Using subtractive notation with ordered pairs:
          // [(1000,'m'), (900,'cm'), (500,'d'), (400,'cd'),
          //  (100,'c'), (90,'xc'), (50,'l'), (40,'xl'),
          //  (10,'x'), (9,'ix'), (5,'v'), (4,'iv'), (1,'i')]
          a. WHILE number >= numeral_pair.value:
              i. ADD numeral_pair.symbol TO result
              ii. SUBTRACT numeral_pair.value FROM number
      4. TRANSFORM result TO lowercase
      5. RETURN result

    POSTCONDITIONS:
      - result MATCHES ^[ivxlcdm]+\$
      - result represents a valid Roman numeral using subtractive notation
  }

  ERRORS {
    INVALID_RANGE: Input number must be between 1 and 1000 inclusive
    HANDLING:
      - INVALID_RANGE: Raise ValueError with message
  }

  EXAMPLES {
    VALID:
      Input: number = 19
      Output: "xix"       // Uses subtractive notation (not "xviiii")

      Input: number = 152
      Output: "clii"

      Input: number = 426
      Output: "cdxxvi"

    INVALID:
      Input: number = 0
      Error: INVALID_RANGE

      Input: number = 1001
      Error: INVALID_RANGE
```

```
  }
 }
```

## A.3   Prompt Templates

### A.3.1   SpecDesigner (Self-Spec language authoring)

```
[SYSTEM]

Create and define an intermediate specification language that you, as the model,
↪   consider maximally readable and reliable for your own use in the NL -> Spec
↪   <-> NL confirmation -> Code pipeline. Use any notation you prefer.

[INSTRUCTIONS]
Make the definition clear enough that another copy of you can (a) write specs in
↪   it from arbitrary NL requests and (b) accurately translate the specs back
↪   into NL (c) implement code from such specs after user confirmation.

[OUTPUT]

Return ONLY the specification.
```

### A.3.2   FMInterviewer (formal clarification Q&A)

```
[SYSTEM]
You are ''FMInterviewer''. From the CURRENT_SPEC (in your invented language), the
↪   ORIGINAL_NL_REQUIREMENTS, and the FUNCTION_SIGNATURE, ask the *fewest and
↪   most critical* questions needed to resolve ambiguities before implementation.

[GOAL]
Elicit only decisions that materially affect correctness, safety, or user
↪   expectations. Default to safe/common assumptions when possible. Avoid asking
↪   cosmetic or trivial questions.

[OUTPUT STYLE]
- Ask at most 5 numbered questions that correspond ONLY to fields still set to
↪   `TBD` in CURRENT_SPEC.
- Each question ends with [default=VALUE]
- Tag each with [<spec.path>], e.g., [Semantics.relation_rule]

[INPUTS]
- CURRENT_SPEC:
{{CURRENT_SPEC}}

- ORIGINAL_NL_REQUIREMENTS:
{{USER_REQUIREMENTS}}

- USER_EXPLANATION (optional, if provided):
{{USER_EXPLANATION}}

- FUNCTION_SIGNATURE:
{{FUNCTION_SIGNATURE}}
```

### A.3.3   SpecInstantiator (schema → instance)

```
[SYSTEM] You are ''SpecInstantiator''.
Fill the SCHEMA with concrete values from the ORIGINAL_NL_REQUIREMENTS and
↪   FUNCTION_SIGNATURE.
If a field is unknown, set it to `TBD`.

[INPUTS]
SCHEMA:
{SCHEMA}

ORIGINAL_NL_REQUIREMENTS:
```

```
{USER_REQUIREMENTS}

FUNCTION_SIGNATURE:
{FUNCTION_SIGNATURE}

[OUTPUT]
Return ONLY the instantiated spec (same format as SCHEMA).
```

### A.3.4   SpecApplier (apply user answers back to spec)

```
[SYSTEM]
You are ''SpecApplier''. Update the CURRENT_SPEC to reflect the user's answers to
↪   the FM questions.

[INSTRUCTIONS]
- Align each user answer with the internal tags you previously emitted (e.g.,
↪   [roles.scope], [concurrency.mode]).
- Rewrite the spec fully in the SAME invented language and style - coherent,
↪   correct, and ready for implementation.
- Preserve all valid prior content; modify only where answers indicate changes.
- Resolve ambiguities by applying the user's decisions; if an answer is missing
↪   or unclear, fall back to the suggested safe default for that tag.
- Do NOT output code.

[INPUTS]
- CURRENT_SPEC:
{{CURRENT_SPEC}}
- Your FM questions (for reference):
{{LLM_questions}}
- USER_ANSWERS (free-text, referencing your tags where possible):
{{USER_ANSWERS}}
- FUNCTION_SIGNATURE:
{{FUNCTION_SIGNATURE}}

[OUTPUT]
Return ONLY the updated spec (in your invented language).
```

### A.3.5   FMConfirmer (natural-language confirmation)

```
[SYSTEM]
You are ''FMConfirmer''. Summarize the key decisions just made, in clear natural
↪   language, suitable for user confirmation.

[INSTRUCTIONS]
- 5-7 sentences max.
- If relevant in the spec, briefly cover: who is allowed to do it; when it is
↪   allowed (states/timing); how concurrent edits are resolved; whether repeated
↪   requests are idempotent; what is recorded/audited/notified; how errors are
↪   communicated; whether policy recalculation or similar rules apply.
- End with: ''Please confirm or tell me what to change. If confirm, simply return
↪   CONFIRM, if you want to change, do not write any code, just tell what to
↪   change to meet with your requirement.''

[INPUTS]
- UPDATED_SPEC (your invented language):
{{UPDATED_SPEC}}
- FUNCTION_SIGNATURE:
{{FUNCTION_SIGNATURE}}

[OUTPUT]
A single natural-language paragraph.
```

### A.3.6   SpecExecutor (implementation from agreed spec)

```
[SYSTEM]
```

```
You are ''SpecExecutor''. Implement the agreed functionality described by the
↪  AGREED_SPEC.
- Deliverables in ONE response:
1) The complete implementation using exactly the provided FUNCTION_SIGNATURE (no
↪  modifications)
2) Ensure the implementation fully satisfies AGREED_SPEC; verify before
↪  outputting.
3) No explanations, no markdown fences.

[INSTRUCTIONS]
- Target language/runtime: {{TARGET_LANG}} {{RUNTIME_VERSION}}.

[INPUTS]
- AGREED_SPEC (your invented language):
{{AGREED_SPEC}}

- FUNCTION_SIGNATURE:
{{FUNCTION_SIGNATURE}}

[OUTPUT]
Only Code.
```

### A.3.7  AgentAnswerer (relay FM questions to user)

```
You requested the following task:
"{{USER_REQUIREMENTS}}"

- FUNCTION_SIGNATURE:
{{FUNCTION_SIGNATURE}}

To make sure everything is clear before implementation, the assistant has drafted
↪  a specification and now asks you to clarify some key points.

Here are the questions:
{{LLM_questions}}

Please reflect on your original request above and answer each question in natural
↪  language.
Do not worry about any internal symbols or notation from the specification - just
↪  give plain, clear answers.
```

### A.3.8  UserConfirmation (final confirmation prompt)

```
Here is the proposed Plan:
{{FINAL_PLAN}}

Here is the original user request:
"{{USER_REQUEST}}"

- FUNCTION_SIGNATURE:
{{FUNCTION_SIGNATURE}}

Please check: Does the Plan fully satisfy the request?

Please confirm or tell me what to change.
If confirm, simply return CONFIRM (exactly this single word).
If you want to change (incorrect implementation), tell what to change to meet
↪  your requirement.
```


