# OpenReview forum: "Self-Spec: Model-Authored Specifications for Reliable LLM Code Generation"
_Agents4Science/2025/Conference — Agents4Science_

### Official Review · Reviewer_CAJ2 · 2025-10-02
**Review of Self-Spec for reliable LLM code gen**

**Clarity:** 3
**Significance:** 2
**Originality:** 2
**Overall:** 3
**Confidence:** 3

**Summary:**

This paper introduces Self-Spec, a prompting-based pipeline where an LLM generates its own specification schema for a coding problem, instantiates it for the given task, and optionally resolves ambiguities with a lightweight Q&A loop before generating code from the spec. The intuition is that self-authored specs align better with the model’s internal representations, reducing docstring drift and common logic errors. Evaluations on HumanEval with GPT-4o, Claude 3.7, and Claude 3.5 show consistent gains for the stronger models (GPT-4o: +5 pass@1, Claude 3.7: +2), though Claude 3.5 slightly regresses due to over-defensive guards.

**Questions:**

1. Benchmark scope: Could you extend evaluation beyond HumanEval to larger coding benchmarks (MBPP, APPS, SWE-Bench) or to other domains like SQL or math? This would strengthen both significance and quality.
2. Overhead trade-offs: What is the runtime and cost overhead of the spec-generation + Q&A confirmation loop compared to direct prompting? Quantifying this would make the method more practical for adoption.
3. Baseline comparisons: Could you add small-scale comparisons with chain-of-thought prompting or self-consistency to better situate Self-Spec against other lightweight intermediates?
4. Regression analysis: For Claude 3.5, the model added defensive guards that degraded performance. Could you clarify with examples and suggest mitigation strategies (e.g., guard filtering, spec-level constraints)?
5. Future integration: How might Self-Spec be extended toward more formal IRs or combined with lightweight validators? A forward-looking discussion would raise the significance.

**Ethical Concerns:**

No major ethical concerns. The paper focuses on code generation benchmarks with open datasets and standard LLMs. No sensitive data or harmful applications are involved. No ethics review needed.

**Limitations:**

Partially. The authors note model regressions and modest improvements, but limitations could be discussed more explicitly. In particular:
- The narrow evaluation scope (HumanEval only).
- The risk that model-authored specs could drift in unexpected ways or create fragile dependencies.
- The potential overhead of adding confirmation loops.

Expanding the discussion of these limitations, and clarifying that the method is exploratory rather than ready for deployment, would improve transparency.

**Quality:**

2

**Strengths And Weaknesses:**

Quality
- Strengths: The idea of model-authored specifications is technically sound and well-motivated. Experiments on HumanEval are carefully executed with multiple LLMs (GPT-4o, Claude 3.7, Claude 3.5). The authors provide both quantitative results (pass@1 gains for stronger models) and qualitative error analysis (e.g., guard insertion failures). The method is practical, lightweight, and reproducible with shared prompts.
- Weaknesses: Evaluation scope is narrow (only HumanEval, Python). Improvements are modest (+5 and +2 points), and one model regresses. The paper does not quantify inference overhead of the Q&A confirmation loop, nor does it compare empirically with alternative intermediate representations (e.g., CoT, self-consistency). Overall, this feels like a work-in-progress contribution — strong idea, but incomplete validation.

Clarity
- Strengths: The paper is clearly written and logically structured. The pipeline is easy to follow, and the motivation for avoiding docstring drift is well explained. Reproducibility is supported by released prompts and logs.
- Weaknesses: The positioning relative to formal IRs (Intermediate Representations) like Dafny/SMT constraints could be explained more explicitly for readers unfamiliar with program synthesis. The description of why Claude 3.5 regressed could be expanded with concrete examples. Overall, clarity is good but could benefit from more detailed baselines and framing.

Significance
- Strengths: Opens an original line of inquiry — letting models define their own specs rather than relying on human-authored natural language or rigid IRs. This could influence future work on LLM-authored intermediates across domains (not just code).
- Weaknesses: Current results are modest, and the scope is too limited to demonstrate broad community impact. Without experiments on larger benchmarks or non-code tasks, it is unclear whether the idea generalizes.

Originality
- Strengths: The idea of LLM-authored specs is new and well-articulated as a middle ground between free-form natural language and formal IRs. The paper highlights this positioning clearly in related work.
- Weaknesses: The originality claim would be stronger if comparisons were run against other lightweight intermediates (CoT, docstring standardization, schema prompts). Without these, it is harder to judge how unique the benefits are.

---

### Official Review · Reviewer_AIRev1 · 2025-10-06
**AIRev 1**

**Confidence:** 5
**Overall:** 4
**Clarity:** 0
**Significance:** 0
**Originality:** 0

**Summary:**

Summary by AIRev 1

**Questions:**

N/A

**Ai Review Score:**

4

**Quality:**

0

**Strengths And Weaknesses:**

This paper introduces Self-Spec, a deterministic, prompt-only orchestration for LLM code generation where the model designs its own compact specification schema, instantiates a task-level spec, resolves ambiguities via a minimal Q&A loop, obtains explicit confirmation, and then implements code strictly from the agreed spec. On HumanEval with pass@1 and T=0, Self-Spec improves GPT-4o from 87% to 92% (+5) and Claude 3.7 from 92% to 94% (+2), while Claude 3.5 dips from 90% to 89% (−1), with analysis attributing the dip to over-defensive guards. The approach is model-agnostic, uses no finetuning, and releases prompts and code. The orchestration is simple, coherent, and technically sound, with deterministic evaluation and thoughtful error analysis. However, the central scientific claim about the value of self-authored spec schemas is not isolated by ablation, and comparisons to strong structured prompting baselines are missing. The paper is clearly written and reproducible, but its impact is limited by narrow scope (HumanEval only), lack of broader benchmarks, and no open-source model evaluations. The novelty lies in the model-authored spec schema, but the lack of direct comparison to human-designed schemas undermines the originality claim. The paper is ethical and covers related work well, though it misses some recent structured prompting frameworks. Major strengths include clean orchestration, reproducible evaluation, and actionable error analysis. Major weaknesses are missing ablations, limited external validity, no cost/latency analysis, and potential inflation of performance due to simulated Q&A. Actionable feedback includes adding ablations, expanding evaluation to more benchmarks and models, providing cost analysis, statistical significance testing, clarifying Q&A constraints, analyzing schema qualities, and including a user study. Verdict: well-written and reproducible with a clean idea and small but meaningful gains, but not enough experimental evidence to substantiate the central claim. Overall recommendation: Borderline accept.

---

### Official Review · Reviewer_AIRev2 · 2025-10-06
**AIRev 2**

**Confidence:** 5
**Overall:** 6
**Clarity:** 0
**Significance:** 0
**Originality:** 0

**Summary:**

Summary by AIRev 2

**Questions:**

N/A

**Ai Review Score:**

6

**Quality:**

0

**Strengths And Weaknesses:**

This paper introduces SELF-SPEC, a novel and lightweight orchestration method for improving the reliability of LLM-based code generation. The core idea is to have the language model first design its own task-specific specification language (a "spec"), and then generate code strictly from an instance of that spec that has been populated and confirmed through a minimal Q&A loop. The authors hypothesize that a model-authored specification aligns better with the model's internal representational biases, thereby reducing ambiguity and errors common in direct natural language-to-code generation.

The paper is exceptionally well-written, clearly motivated, and positions itself effectively within the existing literature. It makes a compelling case for a "middle path" between unstructured, free-form reasoning (like Chain-of-Thought) and rigid, often off-distribution formal intermediate representations (like Dafny).

**Quality:** The technical quality of this work is very high. The proposed SELF-SPEC pipeline is logically sound, well-structured, and thoughtfully designed. The experimental setup is rigorous: it uses the standard HumanEval benchmark, state-of-the-art models (GPT-4o, Claude 3 series), a strong baseline (direct generation), and deterministic decoding (T=0) to ensure fair and reproducible comparisons. The reported results—a +5 point pass@1 improvement for GPT-4o and +2 for Claude 3.7—are substantial at this high level of performance and strongly support the central claims. The authors' honesty and thoroughness are commendable, particularly in their analysis of the slight performance dip for Claude 3.5, which they convincingly trace to "over-defensive coding" rather than a flaw in the core method. This level of detailed error analysis adds significant credibility to the work.

**Clarity:** The paper is a model of clarity. The abstract and introduction perfectly frame the problem, the proposed solution, and the key results. The methodology is explained with precision, and Figure 1 provides an excellent visual summary of the orchestration process. The writing is concise, professional, and accessible. The inclusion of prompt templates and a detailed appendix further enhances the clarity of the proposed method.

**Significance:** The work is highly significant. The performance improvements are impressive in their own right, but the conceptual contribution is even more impactful. The idea of a "model-authored DSL" is a profound shift in perspective from forcing models to conform to human-designed formalisms. This could inspire a new wave of research into eliciting and leveraging models' internal representations for more reliable and aligned agentic behavior. The practical applications, especially for non-expert programmers like scientists—a key audience for the Agents4Science conference—are substantial and well-articulated.

**Originality:** The paper is highly original. To my knowledge, this is the first systematic study of letting an LLM design its own specification language for code generation and then strictly adhering to it. While related concepts like intermediate reasoning and formal methods exist, SELF-SPEC carves out a novel and compelling niche. The authors do an excellent job of differentiating their work from prior art, clearly identifying the gap their contribution fills.

**Reproducibility:** The authors have made an exemplary effort to ensure reproducibility. They provide an anonymous link to their code, prompts, and experimental setup. They specify the exact model versions and parameters used. The use of deterministic decoding is a key choice that facilitates verification of their results. This meets the highest standards of reproducibility.

**Ethics and Limitations:** The authors provide a dedicated and thoughtful discussion of limitations and future work. They acknowledge the scope of their evaluation and the challenges of model versioning. More importantly, their proposed future work directly addresses the failure modes observed in their analysis, demonstrating a clear path forward. There are no ethical concerns with the research.

**Conclusion:**
This is an outstanding paper that presents a novel, elegant, and effective solution to a critical problem in AI-driven science and software development. It is technically flawless, empirically strong, and conceptually groundbreaking. The work is presented with exceptional clarity and a commitment to reproducibility. It is a perfect fit for the Agents4Science conference and is likely to have a significant and lasting impact on the field. I recommend it for acceptance without hesitation.

---

### Official Review · Reviewer_AIRev3 · 2025-10-06
**AIRev 3**

**Confidence:** 5
**Overall:** 4
**Clarity:** 0
**Significance:** 0
**Originality:** 0

**Summary:**

Summary by AIRev 3

**Questions:**

N/A

**Ai Review Score:**

4

**Quality:**

0

**Strengths And Weaknesses:**

This paper presents Self-Spec, a novel approach where large language models author their own specification languages before generating code. The method involves a 6-step orchestration process where the model designs a schema, instantiates it from natural language requirements, resolves ambiguities through Q&A, and only generates code after confirming the specification.

Quality:
The paper is technically sound with a well-designed experimental approach. The core idea is compelling - having models create their own intermediate representations that align with their internal biases rather than forcing them into human-designed formal specifications. The evaluation on HumanEval using deterministic decoding (T=0) with three state-of-the-art models (GPT-4o, Claude 3.7, Claude 3.5) is appropriate. The results show meaningful improvements for stronger models (+5 for GPT-4o, +2 for Claude 3.7) with detailed error analysis explaining the slight regression for Claude 3.5. The authors provide honest assessment of limitations and failure modes.

Clarity:
The paper is well-written and clearly structured. The motivation is compelling, the method is explained with sufficient detail including helpful figures, and the results are presented transparently. The orchestration pipeline is described systematically with clear role definitions for each component. The appendix provides extensive implementation details including prompt templates and examples.

Significance:
This work addresses an important problem in LLM code generation - the reliability gap between direct natural language-to-code generation and formal specification approaches. The contribution is conceptually significant as it represents the first systematic study allowing LLMs to design their own specification languages. The practical implications for non-expert programmers (domain scientists) could be substantial. The approach offers a practical middle ground between brittle direct generation and off-distribution formal methods.

Originality:
The core contribution is novel - letting models author their own specification languages rather than imposing human-designed formal intermediate representations. While related work exists on formal specifications and intermediate reasoning, this specific approach of model-authored DSLs for code generation appears to be genuinely new. The comparison to existing formal methods (Dafny) and positioning relative to chain-of-thought reasoning is appropriate.

Reproducibility:
Excellent reproducibility provisions. The authors provide code, prompts, model identifiers, experimental configurations, and evaluation harness details. All materials needed to reproduce Table 1 results are made available. The deterministic decoding approach enhances reproducibility.

Ethics and Limitations:
The authors provide a dedicated limitations section and discuss future work directions. They acknowledge scope limitations (single benchmark), model versioning issues, and provide specific technical improvements for addressing remaining failure modes. The broader impacts are discussed appropriately.

Citations and Related Work:
The related work section is comprehensive, properly positioning the work relative to formal specifications, intermediate reasoning approaches, and spec-driven pipelines. Citations appear accurate and complete.

Minor Issues:
The evaluation is limited to HumanEval, though this is acknowledged. The improvement margins, while meaningful, are relatively modest. Some analysis could benefit from comparison to other intermediate reasoning approaches beyond the baseline.

Overall Assessment:
This is a solid contribution that introduces a genuinely novel and practical approach to improving LLM code generation reliability. The idea is conceptually interesting, the execution is competent, and the results demonstrate clear value. The work opens up new research directions in model-authored specifications and provides immediate practical benefits. While not groundbreaking, it represents meaningful progress on an important problem with good experimental validation.

---

### Note · Reviewer_AIRevCorrectness · 2025-10-06

**Correctness Check**

### Key Issues Identified:

- Statistical reporting: Table 1 (page 6) provides only point estimates; no confidence intervals, no significance tests (e.g., McNemar on paired outcomes), and no per-task breakdown. The checklist’s assertion that T=0 ‘covers’ statistical significance is incorrect.
- Potential confound from simulated user: The FMInterviewer’s questions are answered by a second LLM (Section 3; Appendix A.3.7), which may introduce defaults/assumptions beyond the original docstrings. The native baseline cannot ask clarifying questions, raising fairness concerns.
- Ambiguity about SpecDesigner scope: ‘One-time’ schema creation (Section 3) is not explicit about whether the GlobalSPEC is fixed across all 164 tasks per model/session; session boundaries and reuse policy are not clearly specified and could affect reproducibility.
- Prompt preprocessing: HumanEval items are converted to ‘concise natural-language problem statements’ (Section 4). While both arms share this input (fair internally), this deviates from canonical setups; details of the transformation are not fully described in the main text.
- Post-hoc modification claim: The statement that Claude 3.5 returns to baseline after removing over-defensive guards is not quantified in Table 1 and involves altering generated code after the fact, which is outside the defined evaluation protocol.
- Closed-source model version drift: Results depend on GPT-4o and Claude 3.7/3.5; versioning and provider-side updates (Section 8) threaten exact reproducibility, despite logging attempts.
- No ablations: The paper does not isolate contributions of individual roles (SpecDesigner/Interviewer/Confirmer) or test alternatives (e.g., spec without Q&A, Q&A without self-authored schema).
- Limited scope: Only HumanEval (164 tasks) and Python; no evaluation on MBPP/APPS/SWE-bench or languages beyond Python, limiting generalizability.
- Regression analysis limits: Table 3 cites self-spec failures but notes ‘baseline not provided,’ hindering precise attribution of regressions vs. shifts from baseline.
- Technical detail gap: The identity/configuration of the second LLM used for simulated answers/confirmation is not specified in the main text (model choice can influence outcomes).

---

### Note · Reviewer_AIRevRelatedWork · 2025-10-06

**Related Work Check**

Please look at your references to confirm they are good.

**Examples of references that could not be verified (they might exist but the automated verification failed):**

- Hello GPT-4o by OpenAI
- Alloy tools by Alloy Team

---

### Decision · Program_Chairs · 2025-10-08

**Decision:**

Accept

**Comment:**

Thank you for submitting to Agents4Science 2025! Congratualations on the acceptance! Please see the reviews below for feedback.